# Health Literacy and Exercise to Treat Frailty in Community-Dwelling Older Adults: A National Survey Study

**DOI:** 10.3390/ijerph19148711

**Published:** 2022-07-17

**Authors:** Chia-Hui Wang, Wen-Pei Chang, Su-Ru Chen, Wan-Ju Cheng, Kuei-Ru Chou, Li-Chung Pien

**Affiliations:** 1School of Nursing, College of Nursing, Taipei Medical University, Taipei 110301, Taiwan; wangch@tmu.edu.tw (C.-H.W.); 10479@s.tmu.edu.tw (W.-P.C.); kueiru@tmu.edu.tw (K.-R.C.); 2Department of Nursing, Shuang Ho Hospital, Taipei Medical University, New Taipei City 235041, Taiwan; 3Post-Baccalaureate Program in Nursing, College of Nursing, Taipei Medical University, Taipei 110301, Taiwan; suru@tmu.edu.tw; 4Department of Psychiatry, China Medical University Hospital, Taichung 404332, Taiwan; s871065@gmail.com; 5Department of Public Health, China Medical University, Taichung 406040, Taiwan; 6Center for Drug Abuse and Addiction, China Medical University Hospital, China Medical University, Taichung 404332, Taiwan; 7Center for Nursing and Healthcare Research in Clinical Practice Application, Wan Fang Hospital, Taipei Medical University, Taipei 116079, Taiwan; 8Psychiatric Research Center, Taipei Medical University Hospital, Taipei 110301, Taiwan; 9Neuroscience Research Center, Taipei Medical University, Taipei 110301, Taiwan; 10Psychiatric Research Center, Wan Fang Hospital, Taipei Medical University, Taipei 116079, Taiwan

**Keywords:** health literacy, exercise, frailty, older adults, national survey

## Abstract

Aging is a major challenge facing modern society and has attracted global attention. Studies have provided some initial evidence that health literacy plays a role in determining frailty; however, most of these studies have used small convenience samples of individuals recruited from geographically limited areas, thus limiting the generalizability of their findings. The present study explored the relationships among health literacy, exercise, and frailty in Taiwanese older adults by using the data of a national population-based survey. We retrieved data from the Taiwan Longitudinal Study on Aging, a population-based survey. We gathered the 2015 data on the age, sex, education level, marital status, exercise habits, and activities of daily living (ADLs) of each eligible respondent. We evaluated the respondents’ health literacy by using a nine-item health literacy scale and categorized their health literacy level as low, medium, or high. Frailty was diagnosed according the Fried criteria. Our final sample consisted of 7702 community-dwelling older adults (3630 men and 4072 adults). Of these, 25.3% had low health literacy. The proportion of respondents who had two or more disabilities in terms of ADLs or instrumental ADLs was higher among the women (36.4% and 12.6%, respectively), and regular exercise was more common among the men (19.6%). Frailty was more prevalent among the women; the prevalence of frailty among the male and female respondents was 4.5% and 8.1%, respectively. High health literacy and regular exercise were protective factors for frailty. According to our results, poor health literacy is a risk factor for prefrailty and frailty, and regular exercise is significantly negatively associated with prefrailty and frailty. Additional studies are necessary to define practical strategies for reducing the risks of disability and death for older adults with low health literacy who do not exercise regularly, thereby improving their quality of life.

## 1. Background

Aging is a major challenge facing modern society, and has attracted global attention. As population aging has developed into a global phenomenon, the rapidly increasing number of older adults experiencing frailty and the economic burden frailty imposes have become major problems [1]. Frailty is associated with high risk of several adverse outcomes, such as cardiovascular and pulmonary diseases, diabetes mellitus, cancers, and depressive symptoms [2,3,4]. The incidences of frailty and prefrailty were reported to be 43.4 and 150.6 per 1000 person-years, respectively [5]. The prevalence of frailty in Japan was reported to be 7.4% among community-dwelling adults older than 65 years, with the prevalence being 8.1% for women and 7.6% for men [6]. The prevalence of frailty was 15% and 4.9% among older non-nursing home populations in the United States and Taiwan, respectively [2,3].

Frailty is a multidimensional syndrome in older adults and is caused by overt age-associated dysregulation of multiple homeostatic systems, leading to the progressive loss of physiological reserves, which may be so severe that it results in multifunctional impairment and eventually in pathology and death. Frailty involves interactions between biological, psychological, and social factors and is difficult to diagnose because of its coexistence with other age-related conditions as well as the lack of a universally accepted clinical definition. The development of effective assessments and intervention strategies for frailty is therefore crucial in aging societies. 

Interventions targeting the dynamic transitions among specific frailty levels may delay the progression of frailty. The phenotype model suggests that five factors (weight loss, self-reported exhaustion, low energy expenditure, slow gait, and low grip strength) are associated with frailty [4,7]. Sarcopenia [8], low-grade inflammation, and deficient endocrine system activity [9] are potential mechanisms involved in the development of frailty. Frailty is also associated with a wide range of sociodemographic characteristics (e.g., age and sex), physical health conditions, and health behaviors as well as exercise interventions [10]. Low health literacy, in cross-sectional observational studies, has been found to be associated with a lack of exercise habits and frailty [11,12]. Shin et al. [12] discovered that after adjustment for demographic and health-related factors, limited health literacy significantly increased the risks of prefrailty and frailty for Korean adults aged 70–84. However, Shah et al. [13] reported that after adjustment for covariates, health literacy was not associated with frailty, and neither were the objective nor subjective numeracy of veteran males (mean age = 56.8 years). However, their sample was a small convenience sample of people recruited from a geographically limited area. Therefore, the generalizability of their findings is limited, and whether low health literacy, low physical activity, and certain sociodemographic characteristics lead to increased risk of frailty among older adults remains unclear. The present study explored the relationship between frailty and health literacy distribution and evaluated associations of frailty with health literacy, exercise, and sociodemographic characteristics among community-dwelling older adults. We hypothesized that good health literacy and regular exercise were the protective factors for frailty status.

## 2. Methods

### Study Design and Population

The Taiwan Longitudinal Study on Aging (TLSA) survey has been conducted every 3 to 4 years since 1989 by the Taiwanese Health Promotion Administration. Initially, the survey respondents were adults aged 60 years or older that lived in nonindigenous Taiwanese townships [14]. However, because the original sample did not include individuals in mountain villages and the sample size began to decrease due to respondents dying or being lost during tracking, the eighth wave of the TLSA survey added new subjects by expanding the survey area to the whole of Taiwan, and nationally representative individuals older than 50 years and subsequent generations were selected for long-term follow-up [15]. In total, eight waves of the TLSA survey have been completed. In the eighth wave, three-stage systematic sampling was employed to randomly select older adults. Details of the sampling design and data collection methods can be found elsewhere [16,17]. The Taiwan Medical University Ethics Committee (TMU JIRB N201907030) approved this analysis of the data from the eighth wave of the TLSA survey.

## 3. Measurement

### 3.1. Frailty Assessment

According to Fried et al. [18], frailty can be represented by five phenotypic components (weight loss, self-reported exhaustion, low energy expenditure, slow gait, and low grip strength). Several studies on frailty have applied similar frailty measurements [3,19,20]. For example, weight loss can be used to identify individuals with poor appetite [19,21]. In this study, poor appetite was identified based the participant’s response to the Center for Epidemiologic Studies Depression Scale (CES-D) item “I did not feel like eating over the past 1 week.” Potential responses reflected how often the respondents had experienced a poor appetite in the preceding week, with response choices of never, rarely (only 1 day), sometimes (2 to 3 days), and often/always (more than 4 days). Responses of never or rarely were classified as robust, and responses of sometimes, often, or always indicated the weight loss phenotype. Similar classification methods have been adopted in other studies [18,19].

To evaluate exhaustion, two items from the CES-D were used: “I felt that everything I did was an effort” and “I couldn’t get going.” The possible responses to these items were never, rarely (only 1 day), sometimes (2 to 3 days), often, and always (more than 4 days). Responses of never or rarely were classified as indicating robustness, whereas responses of sometimes, often, or always indicated the exhaustion phenotype.

Slow gait was evaluated using a questionnaire item used by Nagi [22]: “Do you have any difficulty walking 200 to 300 m?” A response of yes indicated the slow gait phenotype.

Low grip strength was evaluated using the item “Do you have difficulty in picking up or twisting things using your fingers?” [19,21]. A response of yes indicated the low grip strength phenotype.

Participants who did not exercise or who exercised less than once a week were categorized into the low physical activity phenotype [18,19]. Participants with 0, 1 or 2, and 3 or more of the aforementioned phenotypes were included in the robust, prefrail, and frail groups, respectively.

### 3.2. Health Literacy Assessment

Health literacy was measured using a health literacy scale comprising nine items: (1) “When I see a doctor, I can express or explain my condition”, (2) “When I see a doctor, I can understand the doctor’s explanation of or advice on the condition or medical treatment”, (3) “I can read the instructions on the medication packaging or understand the medication instructions given by the doctor”, (4) “I can follow doctors’ instructions for medication (e.g., frequency and dosage)”, (5) “I can understand information related to disease self-management”, (6) “I can follow medical staff’s instructions to self-manage my disease”, (7) “I eat foods that are healthy”, (8) “I know that I must exercise for more than 30 min three times per week”, and (9) “When I experience stress, I know how to reduce my stress”. The internal consistency reliability of the health literacy scale (Cronbach’s α) was 0.87 [23]. Each item was rated on a 5-point Likert scale, and the scores for the nine items were summed to obtain the total score, with a higher score indicating higher health literacy. The health literacy scores were ranked and categorized into low, medium, and high levels.

### 3.3. Exercise Assessment

In the eighth wave of the TLSA survey, exercise was assessed using three items: the number of exercise sessions per week, the duration of each session, and the level of breathlessness the participant experienced in each session. Using the definition of regular exercise of the Taiwanese Health Promotion Administration, Ministry of Health and Welfare [24], we classified people who exercised more than three times per week, who exercised for at least 30 min in each exercise session, and who reached the level of intensity that caused panting during each session into the regular-exercise group. We classified the remaining participants into the irregular-exercise group.

### 3.4. Activities of Daily Living Assessment 

Six items were used to measure activities of daily living (ADLs). The original scale for measuring ADLs was developed by Katz et al. [25] and assesses the degree of difficulty experienced by a person then they are performing six ADLs including eating, getting in and out of a bed or a chair, walking indoors, and going to the toilet without assistance. The Cronbach’s α of the scale has ranged between 0.87 and 0.94 [26,27]. The six items were scored from 0 to 3, with 0 points indicating not difficult, 1 point indicating somewhat difficult, 2 points indicating very difficult, and 3 points indicating impossible. A higher score indicated a higher level of disability. We divided the degree of disability into three categories: no disability, one disability, and two or more disabilities.

### 3.5. Instrumental ADLs

Lawton and Brody [28] developed the scale of instrumental ADLs (IADLs) for evaluating individuals’ abilities to use daily instruments for self-care. The original IADL scale comprised eight items. In this study, five items were used to measure IADLs (buying daily necessities, managing finances, taking transportation, doing housework, and making phone calls). The Cronbach’s α of the Chinese version of the IADL scale ranged between 0.72 and 0.87 [29]. Items were scored from 0 to 3, with 0 indicating not difficult, 1 indicating somewhat difficult, 2 indicating very difficult, and 3 indicating impossible. A higher score indicated more severe disability in terms of IADLs. To calculate descriptive statistics, we divided the degree of disability into three categories: no disability, one disability, and two or more disabilities. Because ADLs and IADLs have a high degree of collinearity, IADLs were excluded from the multivariable logistic regression model.

## 4. Demographic Characteristics

Data on age, educational level, and marital status were obtained through interviews performed by a trained interviewer. However, because educational level and health literacy have a high degree of collinearity, educational level was excluded from the multivariable logistic regression model.

## 5. Statistical Analysis

We performed descriptive analyses of health literacy, ADLs, IADLs, exercise, frailty, and demographic characteristics stratified by sex. The chi-square test was employed to identify differences in categorical variables between the exercise groups. To identify associations of the frailty risk with demographic characteristics, ADLs, exercise, and health literacy, we established a multivariable logistic regression model in which frailty was the dependent variable and sex, age, marital status, health literacy, ADLs, and exercise were independent variables. All analyses were performed using SPSS 24.0 (IBM, Armonk, NY, USA).

## 6. Results

### 6.1. Demographics, Chronic Disease, and Cognitive Impairment Prevalence

A total of 8300 participants completed interviews in the eighth wave of TLSA. In the secondary analysis, we excluded older adults who were living in a long-term care facility or nursing home (*n* = 153) and those whose interview was not conducted in person (*n* = 670). The final study cohort comprised 7702 community-dwelling older people (3630 men and 4072 women). Approximately 25.3% of the older adults had low health literacy. Having two or more disabilities in terms of ADLs or IADLs was much more common among women (36.4% and 12.6%, respectively), whereas a higher proportion of men than women practiced regular exercise (19.6%). Frailty was more prevalent in older women and the prevalence in men and in women were 4.5% and 8.1%, respectively (Table 1).

### 6.2. Frailty and Regular Exercise Distribution in Different Sexes

The community-dwelling older adults of both sexes who reported exercising regularly were more likely to have robust health (86.5% and 77.1% among the men and women, respectively), and had lower prevalence of prefrailty (12.1% and 20.7% for men and women, respectively) and frailty (1.4% and 2.2% for men and women, respectively) than those who reported not exercising regularly (Table 2).

### 6.3. High Health Literacy and Regular Exercise Were Protective Factors for Frailty

The results of the multivariable logistic regression analysis are presented in Table 3. After adjustment for sex, age, and marital status, we found that ADL-related disabilities, health literacy, and regular exercise were significantly associated with frailty. Having two or more ADL-related disabilities was strongly associated with prefrailty and frailty (OR = 1.823, 95% CI = 1.605, 2.071 and OR = 18.826, 95% CI = 13.126, 27.001, respectively). Low health literacy was a risk factor for prefrailty and frailty (OR = 2.734, 95% CI = 2.386, 3.132 and OR = 4.185, 95% CI = 2.961, 5.917, respectively), and regular exercise was significantly negatively associated with prefrailty and frailty (OR = 0.210, 95% CI = 0.179, 0.246 and OR = 0.253, 95% CI = 0.164, 0.393, respectively). Sex and marital status were not associated with frailty, and having two or more ADL-related disabilities was strongly associated with prefrailty and frailty statuses. 

## 7. Discussion

Our results from a national sample of community-dwelling older adults revealed associations between health literacy, exercise, and sociodemographic characteristics with frailty in older adults in Taiwan. Although women have longer life expectancy, women are generally reported to have poorer health outcomes than men. Several studies have revealed sex differences in frailty incidence. Ahrenfeldt [30] reported that European women were more likely than European men to develop common comorbidities (OR = 1.11, 95% CI = 1.07, 1.15) and frailty (OR = 1.56, 95% CI = 1.51, 1.62). Davis et al. [31] conducted a systematic review and meta-analysis of 29 studies involving 8854 adults with heart failure. The results revealed that the women had a 26% higher relative risk of frailty than did the men (95% CI = 1.14, 1.38, z = 4.69, *p* < 0.001). In a longitudinal study, Gale et al. [32] concluded that frailty occurred more frequently in women than in men (16% vs. 12%). Our study also indicated a higher incidence of frailty in older women (8.1%) than in older men. This difference was also reported in Japan [6].

Frailty related to poor health can be considered a state of predisability. Masel et al. [33] discovered negative associations between frailty and various health outcomes in older Mexican-Americans. In another study, frailty could explain differences in 36-Item Short-Form Health Survey scores across all dimensions of the survey; the effects of frailty were stronger than those of chronic physical illnesses such as lower back pain, stroke, and diabetes [34]. In a longitudinal study, Gale et al. [32] reported that frail individuals had mobility problems; physiological changes resulting from frailty can lead to a loss of physical function and dependence on assistance to perform ADLs, eventually leading to hospitalization or extended hospital stays and reducing longevity. Frailty may negatively affect physical, psychological, and social function in older adults through, for example, declines in functional capacity, cardiovascular fitness, depression, and low social participation [35]. In our study, we obtained similar findings; women were more likely to lose two or more ADL or IADL functions than were men.

Exercise reduces age-related oxidative damage and chronic inflammation, increases autophagy, and improves mitochondrial function, myokine profiles, insulin-like growth factor-1 (IGF-1) signaling pathways, and insulin sensitivity. Regular exercise is essential because it improves muscle strength, aerobic capacity, and balance, thereby reducing physical frailty and delaying physical dependence [36]. Strengthening exercises can significantly increase muscle strength and physical mobility in older patients at high risk of sarcopenia or frailty. Numerous studies have demonstrated that exercise increases muscle strength in older adults [36,37,38], increases brain volume (gray and white matter regions and hippocampus) [39,40], and prevents falls [41]. Moreover, exercise improves mobility and physical function in older adults with mobility problems, physical disability, or multiple morbidities [42]. Exercise training is beneficial for older adults regardless of whether they are in a robust condition or have functional impairment. Notably, older adults with frailty are at high risk of multiple comorbidities and adverse events. Regular exercise can effectively decelerate aging and decrease the risk of disabilities in older adults. In our study, frailty was less prevalent in the older adults of both sexes who exercised regularly than in those who did not exercise regularly (5.3% and 9.1% among the men and women, respectively). Regular exercisers also had significantly lower risks of prefrailty and frailty.

Health literacy has been reported to be associated with health behaviors and outcomes in adults with chronic diseases such as obesity, cardiovascular disease, and diabetes [43]. Health literacy has been demonstrated to result in positive information seeking, attitude change, and behavioral change. Moreover, health literacy has been reported to increase motivation and autonomy in the context of adherence to frailty management strategies [12,44]. A study reported that health literacy is an independent predictor of prefrailty and frailty [45]. Health literacy can influence the process of understanding and developing awareness of a disease and may improve the effects of frailty interventions [46]. In addition, health literacy has been identified as a key factor affecting healthcare resource utilization. Inadequate health literacy may prevent patients from utilizing outpatient facilities, leading to increased use of emergency services [47]. Some systemic reviews have revealed that in older adults, low health literacy is associated with more hospitalizations, more frequent use of healthcare services, poorer overall health, and even a higher mortality rate [48,49,50]. Poor health literacy may also be a risk factor for prefrailty and frailty. As discussed above, these results confirm our research hypothesis that good health literacy and regular exercise were the protective factors for frailty status.

## 8. Limitations

Our study has several potential limitations. First, the cross-sectional survey design precluded direct causal inferences. Causal inferences were restricted to the influence of exercise and health literacy on the frailty status of community-dwelling older adults. In addition, reverse causality may occur. For example, socially excluded older adults with frailty may have less opportunity to exercise or to receive new health information to promote their health literacy, thus influencing research inferences. Second, we excluded data provided by proxies to strengthen the validity of the data; however, incorrect reporting may have occurred. Generally, self-reported data are assumed to be reliable and accurate. However, the validity of self-reported data can vary and be influenced by several factors, such as the type of research tool, the wording of items, the respondents’ psychosocial conditions, and recall bias [51]. However, our study was a national survey study, and the sample was sufficiently large to avoid this bias. Third, although the multivariable analysis was performed to adjust for potential confounding factors, additional unmeasured confounding factors may have remained. Finally, our older adults were from Taiwan, which limits the generalizability of the findings to countries with different cultures. In addition, all the studied variables are cross-dependent, and more longitudinal analyses are required to further evaluate the effects of exercise and health literacy on frailty in the future. 

## 9. Conclusions

Our findings, obtained through a national survey, suggest that irregular exercise and low health literacy are associated with increased risk of frailty in older adults. These results have potential implications for policy makers and future studies. The results indicate that it is essential to establish health policies that enable the development and implementation of simple, efficient interventions to prevent frailty and prefrailty in older adults. For example, exercise and health education programs can be included in health management as preventive measures that can be implemented before the onset of old age. Future studies should investigate the extent to which the variables investigated in our study affect frailty in other countries or populations.

## Figures and Tables

**Table 1 ijerph-19-08711-t001:** Demographic characteristics, disabilities in terms of ADLs and IADLs, exercise habits, and Physical condition of respondents of both sexes (*N* = 7702).

Variable	Total		Male (n = 3630)		Female (n = 4072)		*p*-Value
	n/Mean	%/SD	n/Mean	%/SD	n/mean	%/SD	
Age (years)							0.575
50–64	4014	52.1%	1914	52.7%	2030	51.6%	
65–74	2215	28.8%	1027	28.3%	1188	29.2%	
75 & above	1473	19.1%	689	19.0%	784	19.3%	
Education level							<0.001
Primary, can read	3632	47.2%	1372	37.8%	2260	55.5%	
High school and above	4071	52.8%	2259	62.2%	1812	44.5%	
Marital status							<0.001
Married/Cohabitation	5704	74.1%	3053	84.1%	2651	65.1%	
Single/Widowed/Divorced	1998	25.9%	577	15.9%	1421	34.9%	
Health literacy							<0.001
High health literacy	2901	37.7%	1394	38.4%	1507	37.0%	
Medium health literacy	2851	37.0%	1450	39.9%	1401	34.4%	
Low health literacy	1950	25.3%	786	21.7%	1164	28.6%	
ADL							<0.001
No disability	4476	58.1%	2413	66.5%	2063	50.7%	
One disability	934	12.1%	409	11.2%	525	12.9%	
Two disability and more	2292	29.8%	808	22.3%	1484	36.4%	
IADL							<0.001
No disability	6210	80.6%	3115	85.8%	3095	76.0%	
One disability	703	9.1%	241	6.6%	462	11.4%	
Two disability and more	789	10.3%	274	7.6%	515	12.6%	
Exercise							<0.001
Regular	1309	17.0%	711	19.6%	598	14.7%	
Irregular	6393	83.0%	2919	80.4%	3474	85.3%	
Frailty							<0.001
Robust	3907	50.7%	1941	53.5%	1966	48.3%	
Pre-frail	3303	42.9%	1525	42.0%	1778	43.7%	
Frail	492	6.4%	164	4.5%	328	8.1%	

**Table 2 ijerph-19-08711-t002:** Frailty and exercise habits of respondents of both sexes (*N* = 7702).

	Regular Exercise					Irregular Exercise				
Variable	Male (711)		Female (598)		*p*-Value	Male (2919)		Female (3474)		*p*-Value
	n	%	n	%		n	%	n	%	
Frailty condition					<0.001					<0.001
Robust	615	86.5%	461	77.1%		1326	45.4%	1505	43.3%	
Pre-frail	86	12.1%	124	20.7%		1439	49.3%	1654	47.6%	
Frail	10	1.4%	13	2.2%		154	5.3%	315	9.1%	

**Table 3 ijerph-19-08711-t003:** Adjusted odds ratios (ORs) in multivariable logistic regression models of prefrailty and frailty for respondents of both sexes (*N* = 3795).

Variable	Pre-Frail (n = 3303)			Frail (n = 492)		
	OR	(95% CI)		OR	(95% CI)	
Sex						
Male	1			1		
Female	0.023 *	0.890	0.805	1.178	0.949	1.462
Age (years)						
50–64	1			1		
65–74	0.675 ***	0.601	0.759	0.677 **	0.513	0.893
75 & above	0.463 ***	0.397	0.539	0.964	0.743	1.252
Marital status						
Married/Cohabitation	1			1		
Single/Widowed/Divorced	1.100	0.981	1.234	1.038	0.839	1.286
Health literacy						
High health literacy	1			1		
Medium health literacy	2.132 ***	1.903	2.389	2.388 ***	1.677	3.400
Low health literacy	2.734 ***	2.386	3.132	4.185 ***	2.961	5.917
ADL						
No disability	1			1		
One disability	1.111	0.951	1.297	2.566 ***	1.502	4.383
Two disability and more	1.823 ***	1.605	2.071	18.826 ***	13.126	27.001
Exercise						
Irregular	1			1		
Regular	0.210 ***	0.179	0.246	0.253 ***	0.164	0.393

Statically significant at * for *p* < 0.05; ** for *p* < 0.01; *** for *p* < 0.001. Because of the high degrees of collinearity between education level and health literacy and between ADLs and IADLs, education level and IADLs were excluded from the multivariable logistic regression model.

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
