# Peer review of "Health Literacy and Exercise to Treat Frailty in Community-Dwelling Older Adults: A National Survey Study"

_ijerph, 2022, doi:10.3390/ijerph19148711_

Round 1

Reviewer 1 Report

The main strength of the study is the number of studied patients, apart of that there are few key issues, which in my opinion, limit the value of the findings. Please find them listed below. 

1. The study is an observational one and based on a surveys. It is natural that the older people are, they are more likely frail, less active and have a lower health literacy (due to dementia, social exclusion etc.). In other terms, all of the studied variables are cross-dependent, which makes the results quite obvious. In fact, even definition of frailty syndrome includes physical activity! It would be more insightful to analyze activity and healthy literacy in a longitudinal study with frail as an outcome. Currently the study does not provide any new answers. 

2. Statistical methods are poor. There is not multivariable analysis, only single logistic regression. How was it adjusted for age? I do not see it explained? 

3. There are many linguistic errors, which makes the reading hard, only as an example of that please see lines 35-37. 

Reviewer 2 Report

The article presents a relevant and very well-structured theme, according to scientific rules.   With good justification the relevãncia of the problem.   The theoretical framework presents good clarification of concepts, with updated references.   It includes all essential methodological aspects.   The data are presented appropriately and according to the objectives.   Good confrontation of the results with the scientific literature.   As conclusions are consistent with the evidence and arguments presented.

Author Response

Thanks for your comments.

Reviewer 3 Report

The manuscript submitted for review is on the topic: Health Literacy and Exercise to Treat Frailty in Community-Dwelling Older Adults. The topic of the paper appears to be very important from a public health perspective. The topic of health literacy and especially health behaviors of older adults is still a new field in the behavioral research space.

Despite its cognitive value, the paper presents significant shortcomings.

First, the introduction of the paper is very poor in content. I suggest extending it with e.g. the literature on the phenomenon of ageing society and related challenges; moreover, I suggest getting acquainted with the work of Prof. Marcin Duplaga of the Jagiellonian University (e.g. or another scientist who has been working on the issue of health competences for years). At the end of the introduction, please clearly state the purpose of the research and provide research hypotheses/questions that will correspond with the conclusions of the paper.

Methodology section should be re-modeled, in its current form it is unreadable. First of all, I miss clearly presented qualification criteria. I suggest using the layout from other articles appearing under the MDPI publication.

On the editing side, I noticed that Table 3 is repeated at the end of the text (after the references). I do not know why the above procedure serves this purpose.

In addition, the paper uses statistical tests, and in the results themselves, the levels of relationships and their strengths are poorly communicated. Please improve the presentation of the data in the text and provide the test results and p-value.

I wish you all the best!

Round 2

Reviewer 1 Report

Comment number 1 still stands and it cannot be solved. Unfortunately, for this reason I still opt to reject the paper. 

Author Response

  • Comment number 1 still stands and it cannot be solved. Unfortunately, for this reason I still opt to reject the paper.

Author response:

Thanks for your kind comments. I agree with your opinion. Our study was a cross-sectional design, causality cannot be explained, but we hope to identify possible correlation risk factors through this national survey, maybe our future study could focus on the study cohort to identify the causality between regular exercise, health literacy, and frailty status. We revised the introduction section and limitation section. “”Low healthy literacy, although in cross-sectional observational studies, has been found to be associated with a lack of exercise habits and frailty [11, 12 ] (page 2, line 53-54)“ and ““First, the cross-sectional survey design precluded direct causal inferences. Causal inferences were restricted to the influence of exercise and health literacy on frailty status of community-dwelling older adults. In addition, reverse causality may occur. For example, socially excluded older adults with frailty may have less chance to go out to take exercise or receive new health information to promote their health literacy, thus influencing research inferences (page 6, line 260-265)”.

Reviewer 3 Report

The authors have complied with all the comments indicated in the earlier review. I move that the article be accepted in its present form.

Author Response

  • The authors have complied with all the comments indicated in the earlier review. I move that the article be accepted in its present form.

Author response:

Thank you.